# Human Non-Small Cell Lung Cancer-Chicken Embryo Chorioallantoic Membrane Tumor Models for Experimental Cancer Treatments

**DOI:** 10.3390/ijms242015425

**Published:** 2023-10-21

**Authors:** Jing Li, Tereza Brachtlova, Ida H. van der Meulen-Muileman, Stijn Kleerebezem, Chang Liu, Peiyu Li, Victor W. van Beusechem

**Affiliations:** 1Amsterdam UMC Location Vrije Universiteit Amsterdam, Medical Oncology, De Boelelaan 1117, 1081 HV Amsterdam, The Netherlands; 2Cancer Center Amsterdam, Cancer Biology and Immunology, Amsterdam, The Netherlands; 3Amsterdam Institute for Infection and Immunity, Cancer Immunology, Amsterdam, The Netherlands; 4ORCA Therapeutics BV, Onderwijsboulevard 225, 5223 DE ‘s Hertogenbosch, The Netherlands; 5Amsterdam UMC Location Vrije Universiteit Amsterdam, Pulmonary Medicine, De Boelelaan 1117, 1081 HV Amsterdam, The Netherlands

**Keywords:** lung cancer, human xenograft tumor models, tumor growth inhibition, viral vector transduction, gene transfer

## Abstract

To promote the preclinical development of new treatments for non-small cell lung cancer (NSCLC), we established NSCLC xenograft tumor assays on the chorioallantoic membrane (CAM) of chicken embryos. Five NSCLC cell lines were compared for tumor take rate, tumor growth, and embryo survival. Two of these, A549 and H460 CAM tumors, were histologically characterized and tested for susceptibility to systemic chemotherapy and gene delivery using viral vectors. All cell lines were efficiently engrafted with minimal effect on embryo survival. The A549 cells formed slowly growing tumors, with a relatively uniform distribution of cancer cells and stroma cells, while the H460 cells formed large tumors containing mostly proliferating cancer cells in a bed of vascularized connective tissue. Tumor growth was inhibited via systemic treatment with Pemetrexed and Cisplatin, a chemotherapy combination that is often used to treat patients with advanced NSCLC. Lentiviral and adenoviral vectors expressing firefly luciferase transduced NSCLC tumors in vivo. The adenovirus vector yielded more than 100-fold higher luminescence intensities after a single administration than could be achieved with multiple lentiviral vector deliveries. The adenovirus vector also transduced CAM tissue and organs of developing embryos. Adenovirus delivery to tumors was 100–10,000-fold more efficient than to embryo organs. In conclusion, established human NSCLC-CAM tumor models provide convenient in vivo assays to rapidly evaluate new cancer therapies, particularly cancer gene therapies.

## 1. Introduction

Lung cancer is the most common cause of cancer death worldwide, accounting for nearly 1.8 million deaths in 2020 [1]. In addition, lung cancer has the highest disease burden (in disability-adjusted life years) of all cancers [2]. Non-small cell lung cancer (NSCLC) accounts for about 85% of lung cancer cases [3]. Current treatment strategies for NSCLC consist of surgery, radiation, and chemotherapy. Despite efforts for targeted treatment and immune checkpoint inhibition, which have shown clinical effects in a minority of patients, prognosis, particularly in advanced diseases, remains dismal [4]. Thus, there is an urgent need for more effective therapeutic strategies.

Important progress in the treatment of NSCLC is expected particularly from the discovery of new therapeutic targets and cancer-selective drugs and the advancement of gene and cell therapies. To facilitate the development of such new treatments, there is a need for relevant preclinical NSCLC models that allow a quick and reproducible evaluation of the efficacy of novel anti-tumor compounds and treatment regimens. Currently, preclinical research on NSCLC treatment mostly relies on cell culture models and animal models. Cell culture models range in complexity from established cell line models to primary heterogeneous cell cultures and from two-dimensional (2D) monolayer cultures to 3D spheroid or organoid cultures [5]. The most simple 2D cell cultures are widely used in preclinical screening for new targets and compounds, as well as in high-throughput settings, but their limitations are evident. These cultures cannot reproduce the interactions between cells and the extracellular matrix, which easily leads to the dysregulation of cell signaling pathways [6] and thus changes cell biology. In addition, they do not recapitulate the physical barriers to the diffusion of oxygen, nutrients, and experimental compounds that exist in tumors [7]. Though it is generally accepted that 3D cell cultures such as organoids more adequately represent the tumor pathophysiology and more closely mimic the in vivo tumor microenvironment conditions, they are still far from perfect. For example, most 3D systems lack tumor stroma and vascularization. In addition, the extensive use of these models is hampered by their limited reproducibility and reliability and relatively high costs [8,9]. Moreover, 3D cell cultures are usually static, lacking fluid flow and dynamic interactions with other parts of the living body. In contrast, animal models for cancer allow the investigation of novel treatments in the context of a complete organism. A large variety of rodent, mainly mouse, models are used for this purpose [10]. Researchers can either establish syngeneic tumors in immune-competent animals or xenograft tumors in immune-deficient animals. While these models also have their limitations, they are generally considered the most appropriate for the validation of new treatments before clinical translation. However, considering their complexity and high costs and the strong ethical wish to reduce the use of experimental animals, animal models are less preferred to test a multitude of agents, concentrations, or treatment schedules. As such, alternative in vivo tumor models that can bridge initial cell culture experiments to validation in animal models are very useful. Ideally, these models are reproducible and cheap, allow quick testing of treatments, and do not require ethical approval that may delay the initiation of experiments considerably. The chicken chorioallantoic membrane (CAM) xenograft tumor assay fulfills these requirements [11].

Since the early 1970s, the CAM of fertilized chicken eggs has been used as a model for studying neovascularization [12]. The CAM, which primarily serves as the respiratory organ of avian embryos, is formed via the allantois that extends extra-embryonically from the ventral wall of the endodermal hindgut and fuses with the mesodermal layer of the chorion (Figure 1a). This structure that connects to the embryonic circulation expands rapidly from embryonic development day (EDD) 4 to EDD 10 and completes its capillary network development by EDD 14, generating rich vascular networks [13]. The rapid and high vascularization and easy access of the CAM provide a suitable platform and well-nourished environment to establish tumor models [14]. Lymphangiogenesis in the CAM occurs during the mid-development phase, following the initial growth of the blood vascular system, but the immune system reaches physiological activity only by EDD 15, and the chicken embryo does not become immunocompetent before EDD 18 [15,16]. The natural immunodeficiency of the embryo in the early and midstages of development allows the CAM to be widely used as a platform to sustain xenografted tissues and cells. Experience shows that human tumor growth on the CAM is fast. It takes only a few days to grow macroscopically visible tumor tissue on the CAM after the inoculation of human tumor cells [17,18], whereas, in rodent models, this may require up to months [19]. Figure 1b illustrates a typical scheme for a CAM tumor experiment, which is completed within 2.5 weeks. Notably, experiments on fertilized chicken eggs are currently not classified as animal experiments in the European Union countries if they are terminated before hatching and, therefore, do not require animal experimentation committee approval. In addition, the CAM tumor assay is low-cost, particularly compared to experiments in immune-deficient animals. Together, this makes the CAM tumor model readily accessible for the confirmation of the results from in vitro studies before validation in animals. In recent years, CAM models for human tumors were described for a variety of cancers, including, e.g., glioma, osteosarcoma and carcinomas of the colon, ovary, esophagus, prostate, head and neck, and pancreas [17,18,20,21,22,23,24]. Here, we set out to develop tumor models for human NSCLC on the CAM using a panel of cell lines with different driver mutations and to establish their utility in testing experimental treatments for lung cancer. To test established and novel treatments, we used chemotherapeutic agents and lentiviral and adenoviral vector-mediated gene transfer.

## 2. Results

### 2.1. Establishment of Five Different Human NSCLC-CAM Tumor Models

Human NSCLC xenograft tumors were grown on the CAM of fertilized chicken eggs according to the schedule illustrated in Figure 1b. The rapid development of fertilized chicken eggs with associated physiological changes demands strictly standardized procedures for establishing xenograft tumors. The day the eggs were transferred to an incubator set at 37.5 °C was considered EDD 0. On EDD 6, fertilized chicken eggs with viable embryos were selected, and human NSCLC cells in Matrigel suspension were grafted onto the CAM of these eggs. As such, five NSCLC cell lines with different genetic driver mutations, i.e., SW1573, A549, H1299, H292, and H460 cells, were chosen. In pilot experiments, we varied the number of cells in the range of 1 × 10^6^ to 3 × 10^6^, which yielded similar results. Therefore, in subsequent experiments, 1 × 10^6^ cells were always used. On EDD 9, i.e., three days after grafting cells, embryo viability was reassessed, and tumor take rates were documented. Only a few embryos were lost between EDD 6 and EDD 9; on viable embryos, all five human NSCLC cell lines showed good engraftment (above 80%; Table 1). While tumor engraftment was very efficient, the dimensions of the tumors could not be measured in all cases. Some tumors were partially covered by the Matrigel. Figure 2a includes examples of such partially covered tumors. As long as the Matrigel did not obscure the border between tumor and CAM tissue, tumor dimensions could be measured and volumes calculated. For the different NSCLC cell line-derived CAM tumors, in 35–55% of the cases, this was not possible. These tumors were excluded from tumor volume measurements. From EDD 9 to EDD 18, embryo survival and tumor growth were monitored. Embryo survival was high, i.e., 94% across the five NSCLC cell lines at EDD 18 (Table 1). Figure 2 shows representative microscopic images of the NSCLC-CAM models at different time points (Figure 2a) and the calculated average tumor volumes (Figure 2b). Notably, the tumor volume calculation assumes an elliptic tumor shape. As tumors are often irregularly shaped, the calculated volumes should be considered the best estimates. As can be seen, all five models exhibited consistent growth, albeit with variable growth rates. Based on a power analysis, the required sample sizes for reliable detection of treatment effects on the five models were calculated (Table 1). A549 lung adenocarcinoma and H460 large cell lung cancer CAM tumor models, representing one of the two slowest growing NSCLC-CAM tumor models and the fastest growing NSCLC-CAM tumor model, respectively, exhibited the smallest variation in growth rate, hence requiring the smallest group sizes. Therefore, the A549-CAM and H460-CAM models are the preferred choices when performing tumor growth inhibition experiments with multiple treatment groups.

### 2.2. Histological Characteristics of A549-CAM and H460-CAM Tumor Tissues

On embryonic development days 9, 11, 14, and 18, i.e., 3–12 days after inoculating the NSCLC cells, the A549-CAM and H460-CAM tumor tissues were dissected and processed for (immuno)histochemical analysis. General tissue structures were assessed via hematoxylin-eosin staining. Human NSCLC cells were identified via immunostaining for the DNA repair enzyme Apurinic/Apyrimidinic Endodeoxyribonuclease 1 (APE1). APE1 is highly expressed in NSCLC, where it is associated with poor prognosis. Its expression in A549 and H460 cells was previously confirmed [25,26]. Ki-67 staining was used to mark proliferating cells [27]. Vascularization was investigated via staining for Lens culinaris agglutinin (LCA). LCA is one of the few lectins that efficiently binds to the chick embryo vascular endothelium. It was shown to uniformly bind to arteries, capillaries, and veins at different developmental stages of the chicken embryo [28]. Control LCA staining of the CAM tissue from a fertilized egg without transplanted NSCLC cells showed that LCA bound to chicken CAM cells with the most intense staining observed in vessels (Appendix A). Figure 3 shows the histological characteristics of the A549-CAM and H460-CAM tumors. Three days after inoculating the NSCLC cell suspension in Matrigel onto the CAM, NSCLC solid tumors were already established. The two types of NSCLC tumors exhibited quite distinct structures, with A549 cells forming compact tumors, with a relatively uniform distribution of cancer cells and host stroma cells (Figure 3a), and H460 cells forming multiple nodules of cancer cells in a larger bed of highly vascularized connective tissue (Figure 3b). In addition, H460-CAM tumors contained a large fraction of proliferating cancer cells. By scoring the percent Ki-67 positive cells over APE1-positive cells, this was estimated at approximately 100% (65–163% on individual images). In contrast, A549-CAM tumor cells were mostly Ki-67-negative (estimated only 10–64% proliferating NSCLC cells on individual images). These observations are in line with the slow and fast growths measured microscopically for A549-CAM and H460-CAM tumors, respectively. The vascularization of the tumors (at least in the stromal areas of H460-CAM tumors) was already evident from EDD 9 onwards, suggesting that systemic treatments can be tested in these models as early as starting from EDD 9, i.e., only 3 days after inoculating the NSCLC cells onto the CAM.

### 2.3. Chemotherapy Treatment of A549-CAM and H460-CAM Tumors

To investigate the utility of the established NSCLC-CAM tumors to assess the in vivo efficacy of anticancer therapies, we subjected A549-CAM and H460-CAM tumors to systemic treatment with Cisplatin and Pemetrexed. This cytotoxic chemotherapy combination is an important component of systemic treatment for most patients with advanced NSCLC [29]. Since this treatment had not been tested in a CAM tumor model before, we first assessed a safe dose regimen in fertilized chicken eggs. Cisplatin and/or Pemetrexed were administered to the CAM on EDD 9 or 11 at various concentrations, and embryo vitality was monitored from EDD 9 to EDD 18 (Appendix A). The main criteria for vitality were the heartbeat of the embryo, clear and light red blood vessels (indicating the circulation of fresh oxygenated blood), and the presence of small vessels along the shell. Chemotherapy given on EDD 9 was highly toxic, except for the low-dose treatment with Cisplatin only. In contrast, when chemotherapy was delayed until EDD 11, 100% embryo survival was observed in combination treatment concentrations up to 10 mg/kg Pemetrexed with 0.2 mg/kg Cisplatin. Therefore, this dose combination was used to investigate tumor growth inhibition in A549-CAM and H460-CAM models.

In both NSCLC-CAM models, systemic chemotherapy significantly inhibited tumor growth in vivo (Figure 4). Most treated A549-CAM tumors were smaller on EDD 17 than on EDD 11. While treated H460-CAM tumors did still grow, the inhibitory effect of Cisplatin plus Pemetrexed on these fast-growing tumors was more profound. In addition, three treated H460-CAM embryos died, and for three others, tumor sizes could not be assessed on EDD 17 because the limits of the tumors were not clearly visible anymore, possibly as a result of extensive tumor cell lysis. The chemotherapy effect on H460-CAM tumors might thus be underestimated. Hence, both slow-growing A549-CAM and fast-growing H460-CAM tumors were successfully treated with a cytotoxic chemotherapy combination known to be effective in NSCLC, suggesting that both models are useful for the preclinical testing of systemic NSCLC treatments.

### 2.4. Lentiviral Vector-Mediated Gene Delivery into Established NSCLC-CAM Tumors

Since many novel treatments involve the transfer of DNA, e.g., to express a transgene or to silence or edit an endogenous gene, and lentiviral vectors (LVs) are an often used gene delivery tool in preclinical experiments, we investigated the susceptibility of established NSCLC-CAM tumors to LV-mediated gene delivery. For these experiments, we used a lentiviral vector expressing firefly luciferase and monitored transduction via bioluminescence measurement. We first determined the most effective route of LV administration in pilot experiments. Eight days after establishing SW1573-CAM tumors, i.e., on EDD 14, LV-Fluc was administered in a single dose of 1 × 10^6^ TU or 1 × 10^7^ TU. Four delivery routes were compared, i.e., via application onto the CAM surface, direct injection into the tumor, injection into the yolk sac, or injection into the allantoic cavity of the egg. Bioluminescence measurement was performed 48 h later on EDD 16. As seen in Appendix A, only delivery via the CAM presented detectable luminescence in the tumor. Therefore, this practically most convenient administration route was chosen for further experiments.

Next, we compared different LV-FLuc dosing schedules on A549-CAM and H460-CAM tumors (Figure 5 and Appendix A). LV-FLuc was applied onto the CAM of A549 tumor-bearing eggs at doses of 1 × 10^7^, 1 × 10^8^, or 1 × 10^9^ copies on EDD 9, 11, and 13 or at 1 × 10^9^ copies only once on EDD 9. Luminescence signals measured after 24 and 48 h were similar (Appendix A). Generally, luminescence was very low; in many cases, it did not rise above the background (Figure 5). However, if luminescence was detected, this was emitted from tumors, not from any other location, including the LV application site (Figure 5a). Hence, systemic LV-FLuc delivery via the CAM preferentially transduced tumor cells. Although a higher median luminescence signal was observed upon the administration of the highest titer LV-FLuc (Figure 5b), differences between dose groups were not significant (Kruskal–Wallis test). Increasing the LV dose 100-fold (three administrations of 1 × 10^9^ copies versus three times 1 × 10^7^ copies) yielded only a few fold higher median luminescence values. Similar, but slightly lower transduction was seen in H460-CAM tumors (Figure 5a,b). In an attempt to maximize the detection of LV transduction, 1 × 10^9^ LV-FLuc copies were applied onto the CAM of A549-CAM- and H460-CAM-bearing eggs daily from EDD 9 to EDD 13, and tumors were dissected before imaging on EDD 15 (Figure 5c). Again, luminescence was detected, but signal intensities were not higher than was observed upon a single LV-FLuc administration. Overall, only a minority of LV-treated eggs showed convincing luminescence from the NSCLC tumor. Thus, an in vivo tumor transduction via LV-mediated gene transfer was possible but not efficient.

### 2.5. Adenovirus Vector-Mediated Gene Delivery into Established NSCLC-CAM Tumors and Host Chicken Tissues

Because LV-mediated gene delivery was not very efficient, we investigated the feasibility of using an adenovirus vector (AdV) for the delivery of the firefly luciferase gene in the NSCLC-CAM tumor model. In a pilot experiment, we applied AdCMV-Luc in a single dose of 5 × 10^7^ IU to fertilized chicken eggs at EDD 10. The eggs either carried established A549-CAM or H460-CAM tumors or were free of tumors. AdCMV-Luc was administered either via application onto the CAM or intratumoral injection. Bioluminescence was measured on EDD 11, 14, and 17, i.e., 24, 96, and 168 h post-infection (hpi) (Figure 6). Reproducibly, transient luciferase expression was detected, regardless of the route of administration. Luminescence was generally higher in eggs bearing H460 tumors than in eggs with A549 tumors (Figure 6b). Interestingly, luciferase expression was also detected upon the delivery of AdCMV-Luc onto the CAM of eggs without tumor. Thus, the systemic delivery of AdCMV-Luc transduced human NSCLC tumors and chicken tissue. Although AdV- and LV-mediated gene delivery efficiencies cannot be compared directly, average luminescence intensities measured upon a single AdCMV-Luc administration were more than 100-fold higher than could be reached with multiple LV-FLuc additions (Figure 6c).

Because transduction was also seen in CAM tissue, we investigated the biodistribution of AdCMV-Luc in the chicken embryo. We administered AdCMV-Luc to A549- or H460-bearing eggs either via application onto the CAM or intratumoral injection or to tumor-free controls via the CAM. The vector was applied on EDD 10, EDD 12, or EDD 17. Twenty-four hours later, the D-luciferin substrate was administered, and luminescence was recorded. All AdCMV-Luc-treated eggs emitted a luminescence signal (not shown). Subsequently, individual tissues (tumor, CAM, lung, heart, liver, crop, kidney, spleen) were dissected when macroscopically discernable and imaged for bioluminescence. In accordance with the imaging of the intact eggs, luciferase expression was detected in the dissected tumor and/or CAM tissue of all eggs (Figure 7a). Tumor transduction was similarly efficient on the three administration days and using the two administration routes (Figure 7b). Interestingly, two embryos treated on EDD 12 exhibited luminescence in the liver and kidneys, and two embryos treated on EDD 17 had detectable luminescence in the liver and spleen (Figure 7c).

After measuring bioluminescence, tissues were flash-frozen, and DNA was isolated to determine AdCMV-Luc copy numbers via qPCR analysis (Figure 8). This confirmed that vector delivery to tumor and CAM was much more efficient than to embryo organs regardless of the administration route (*p* < 0.05 for all organs versus tumor on at least one delivery day; Kruskal–Wallis test). Copy numbers in tumors were 100–1000-fold higher than in embryo organs. Thus, like LV, systemically administered AdV preferentially transduced tumors. Delivery to the tumor via intratumoral injection appeared to become more effective on EDD 12 and EDD 17 than on EDD 10, although the difference did not reach significance (Kruskal–Wallis test) (Appendix A). Increased transduction at later EDDs was not due to tumor growth, as transduction per mg tissue was similarly increased (Appendix A). We observed a trend that intratumoral injection was more efficient in A549 than in H460 tumors, but this difference did not reach significance. Apart from being much more efficient than LV in transducing NSCLC tumors, AdV also efficiently transduced CAM tissue. On EDD 10, AdV delivery to the CAM was most effective if the vector was delivered topically onto the CAM (*p* < 0.05, Kruskal–Wallis test; Figure 8), but in later days, delivery via intratumoral injection was similarly effective.

Although average AdCMV-Luc copy numbers in organs were several orders of magnitude lower than in tumor and CAM, the vector was detectable showing that embryo organs were reached via systemic delivery. While all tested organs could be transduced via the AdV, differences in delivery efficiency were observed between delivery days and routes. On EDD 10, the most effective way to deliver AdCMV-Luc to embryo organs was to apply the vector onto the CAM of tumor-bearing eggs. Surprisingly, transduction was less efficient if the vector was applied to the CAM of eggs without NSCLC tumor (*p* < 0.05 for liver, lung, and heart, Kruskal–Wallis test; trend for the other tissues; Figure 8), suggesting that tumor growth influenced the capacity of the CAM to take up AdV. In contrast, at later time points, intratumoral injection appeared most effective, possibly because tumors were further developed then, providing a richer vascular bed for systemic delivery. For the liver, lung, crop, and kidney, transduction via intratumoral injection was higher at EDD12 and EDD17 compared to EDD10, and the transduction of the later developing spleen was only evident on EDD17 (Appendix A). Two embryos treated via intratumoral injection on EDD 12 reached copy numbers in the liver and kidney in the range of what was reproducibly achieved in tumor and CAM. These samples were among the exceptional organs with detectable luminescence (Figure 7b). In contrast, when AdCMV-Luc was administered via the CAM, the transduction of the liver and crop was only effective on EDDs 10 and 12 (Figure 8 and Appendix A). Thus, for these organs, intratumoral and CAM AdV delivery routes had opposing transduction patterns.

## 3. Discussion

In this study, we established NSCLC-CAM tumor models that can be used for the in vivo testing of experimental tumor treatments. The ultimate aim is to increase the throughput of testing different treatment conditions to accelerate preclinical development while reducing the use of experimental animals. Recently, several reports were published in which NSCLC-CAM tumors were used to test anticancer treatments. In line with the established use of the CAM model to study neovascularization, two of these studies focused on the antiangiogenic effects of anticancer agents by investigating vascular structure formation [30,31]. The test compounds were either mixed with the NSCLC cells before grafting them on the CAM or were applied during the first few days when vascularized CAM tumors were formed. In another study, NSCLC cells were transfected with siRNA before grafting them to the CAM and tumors were measured to investigate the role of the silenced genes in tumor formation [32]. In yet another report, an experimental anticancer compound was applied topically to the growing tumors for 6 days, and tumor growth inhibition was quantified by weighing the excised tumors [33]. Interestingly, by grafting only very few cells, it was shown to be possible to detect metastatic spread of NSCLC cells in the chicken embryo [34]. When combined with sensitive imaging techniques, this allows for the testing of new treatments for metastatic disease as well. Our aim was to design models to test anticancer treatments on established NSCLC tumors, preferably via systemic delivery. Therefore, we first characterized growing NSCLC-CAM tumors to select the most useful NSCLC cell lines and define a window for experimental treatment. Next, we confirmed the utility of the models using chemotherapeutic drugs with known clinical activity against NSCLC. Finally, with a particular interest in gene therapies, we focused on developing methods for viral gene delivery in established NSCLC-CAM tumors. 

As CAM tumor models were already developed for a variety of human cancers [17,18,20,21,22,23,24], we could use the described procedures for the establishment of human tumors on the CAM that needed only a little adjustment to optimize the formation of NSCLC tumors. All five human NSCLC cell lines included in our studies rapidly formed solid tumors on the CAM, and their growth could be followed until the termination of the experiment. Of these, A549-CAM and H460-CAM tumors grew most reproducibly, with an acceptable variation requiring the smallest group sizes to detect significant effects of anti-cancer treatments. Therefore, these two NSCLC-CAM models were chosen for further studies. The two tumors displayed very different histological and growth characteristics. While A549-CAM tumors exhibited rather homogenously distributed NSCLC cells with a low fraction of proliferating cells and expanded in volume only up to 3.7-fold in 9 days (i.e., from EDD 9 to EDD 18); H460-CAM tumors showed multiple nodules of highly proliferating NSCLC cells embedded in vascularized connective tissue and expanded 20–25-fold in 9 days. The two models are thus very useful to test the efficacy of anticancer treatments on tumors with slow and high growth rates, respectively. Based on the observed tumor vascularization, we concluded that systemic treatments could start as early as three days after NSCLC cell inoculation. This allows a window to treat established NSCLC tumors for more than one week.

Anticancer treatments that could be considered for evaluation in the NSCLC-CAM models include, e.g., chemotherapy, radiotherapy, targeted treatments with small molecule compounds, and gene therapies. The irradiation of CAM tumors is feasible, as this was already carried out successfully to test the combination of radiotherapy with antiangiogenic treatments [20,35]. For most experimental in vivo treatments, it is crucial that compounds are delivered to the tumor growing on the CAM, preferably via systemic administration. When we tested chemotherapeutic drugs and recombinant viruses in NSCLC-CAM models, we observed that the simple application of agents onto the CAM resulted in efficient systemic delivery. The highly vascularized CAM thus not only serves as an excellent substrate to grow xenograft tumors, but it also efficiently takes up chemical molecules and gene delivery vectors that are of interest for cancer therapy development. In the early developmental stage, embryos exhibited high vulnerability to the tested chemotherapeutic agents Pemetrexed and Cisplatin. Doses (in mg/kg) had to be reduced compared to doses that are generally used in mouse studies [36]. The high toxicity of chemotherapy was not unexpected for a developing embryo and suggests that CAM tumor models can also be used for sensitive safety testing of experimental compounds. Systemic treatment of A549-CAM and H460-CAM tumors with a combination of Pemetrexed and Cisplatin at sub-lethal doses effectively inhibited tumor growth in both NSCLC models. This confirmed the relevance of these assays, as they reproduced the anticancer effect of a chemotherapy combination that is commonly used to treat NSCLC [29]. The main limitation of these experiments was the relatively short duration of the monitoring period. This is a generally recognized disadvantage of CAM tumor assays [37]. After treatment with Pemetrexed/Cisplatin, tumor growth could be followed for one week, during which growth inhibition was almost complete. The experiments thus convincingly demonstrated the in vivo efficacy of the tested treatment, but studies in CAM tumor models cannot ascertain durable responses.

Importantly, in view of novel therapy developments, efficient viral vector-mediated gene delivery into NSCLC-CAM tumors was achieved. The systemic delivery of lentiviral or adenoviral vectors expressing the firefly luciferase reporter gene transduced established NSCLC-CAM tumors. Notably, transduction with an AdV yielded more than 100-fold higher luminescence intensities after a single administration than could be achieved with multiple high-dose LV deliveries. In terms of tumor transduction, adenovirus thus clearly outperformed lentivirus. In addition, in contrast to the LV, which only detectably transduced NSCLC-CAM tumors, AdV also efficiently transduced CAM tissue and, albeit much less efficiently, multiple organs of developing embryos. This showed that AdV could reach the intra-embryonic circulation via the chorioallantoic vein. Depending on the EDD, different organs could be transduced in the developing embryo. For example, while the transduction of the embryo liver was seen in the early as well as late stages of embryo development, the developing spleen could only be transduced after EDD 12. This is in line with the immune system development in the chicken embryo where erythropoiesis starts in the spleen from EDD 11 and functionally competent macrophages can be observed in the spleen at EDD 16 [38,39]. Interestingly, the presence of a growing NSCLC tumor increased systemic AdV delivery to embryo organs, and transduction efficiency to some organs changed during embryo development depending on the delivery route. These effects are perhaps caused by changes in the circulatory system.

The much higher gene transfer efficiency observed with AdV than with LV in the CAM tumor model suggests that for many preclinical cancer gene therapy studies, AdV will be the more appropriate vector. Its use will maximize gene expression in CAM tumors, providing the best chance to assess anticancer treatment efficacy and allow the analysis of the safety of therapeutic gene delivery by studying toxicity to non-malignant tissues. Interestingly, while the initially immune-deficient chicken embryo allows for the engraftment of human tumors, it develops a physiologically reactive immune system in the last few days before hatching [15,16,38]. Notably, tumor-bearing embryos were found responsive to immune checkpoint inhibition [40], suggesting that the CAM tumor model could also be used in cancer immunotherapy studies. An obvious limitation of such investigations is the very short existence of established xenograft tumors in an immune-competent environment. The effects of immune-modulating agents can only be monitored in CAM tumor models for a few days. This probably impedes the detection of significant tumor growth inhibition. However, the investigation of immune cell infiltration and activation in tumors seems feasible and meaningful. In conjunction with our findings on adenoviral vector delivery into established NSCLC-CAM tumors, this could perhaps provide an opportunity to test adenovirus vectors expressing immune modulators or oncolytic adenoviruses in human tumor xenograft models.

## 4. Materials and Methods

### 4.1. Cell Culture

A549, SW1573, (NCI-)H1299, (NCI-)H292, and (NCI-)H460 NSCLC cell lines and HEK293T and 911 cells were obtained from the cell line repository of the Laboratory Medical Oncology, Amsterdam UMC, The Netherlands. Known mutations in NSCLC-associated genes in the five NSCLC cell lines were extracted from the Catalogue Of Somatic Mutations In Cancer (COSMIC) database Cell Lines Project v98 (www.sanger.ac.uk/tool/cosmic/, released 23 May 2023 and accessed on 8 September 2023). Appendix A gives the number of currently identified mutations in genes that are causally implicated in cancer according to the Cancer Gene Census and lists mutation details on a selection of genes that are known to be associated with NSCLC. Because the Cancer Gene Census is not static, we also include the COSMIC sample ID for the cell lines to aid in retrieving updated information. NSCLC cell line identity was confirmed via short tandem repeat analysis (outsourced to BaseClear, Leiden, The Netherlands), and all cell lines were tested negative for mycoplasma every 3 months. All cell lines were maintained in Dulbecco’s Modified Eagle’s Medium-high glucose (Sigma-Aldrich Chemie NV, Zwijndrecht, The Netherlands, #D5796) supplemented with 10% Fetal Bovine Serum (FBS; Gibco™, Fisher Scientific, Landsmeer, The Netherlands, #10270-106) and 1% penicillin/streptomycin (P/S; Sigma-Aldrich Chemie NV, Zwijndrecht, The Netherlands, #P4333). During experiments, P/S was omitted from the medium. All culturing procedures were performed at 37 °C with 5% CO_2_.

### 4.2. Viral Vector Production and Titration

Lentiviral vector (LV) expressing firefly luciferase was made by transfecting 3 × 10^6^ HEK293T cells with firefly luciferase expression plasmid Ubc.Luc.IRES.Puro [41] together with pMD2.G and psPAX2 packaging constructs (4 μg mix of three plasmids in 1:4.4:4.7 molar ratio), using FuGENE^®^ HD (Promega Benelux, Leiden, The Netherlands, #E2311) transfection reagent at 3 μL per μg DNA. Ubc.Luc.IRES.Puro (Addgene plasmid, Watertown, MA, USA, # 33307) was a gift from Linzhao Cheng; psPAX2 (Addgene plasmid, Watertown, MA, USA, #12259) and pMD2.G (Addgene plasmid, Watertown, MA, USA, #12260) were gifts from Didier Trono. The next day, the culture medium was changed to DMEM with 30% FBS and 1% P/S. Two days after transfection, the culture medium containing virus particles was harvested and cleared via centrifugation. LV preparations were concentrated using PEG-it™ Virus Precipitation Solution (System Bioscience SBI, Sanbio B.V., Uden, The Netherlands, #LV810A-1). Functional transduction unit (TU) titer was determined based on the capacity to confer puromycin resistance to recipient cells. SW1573 cells plated 3 × 10^4^ per well in 24-well plates were subjected to 10-fold serial dilutions of LV preparations in culture medium with 4 μg/mL polybrene (Sigma-Aldrich Chemie NV, Zwijndrecht, The Netherlands, #TR-1003-G). The next day, the medium was replaced, and one day later, 0.5 μg/mL puromycin (Alfa Aesar, Brunschwig Chemie BV, Amsterdam, The Netherlands, #J67236) was added. Puromycin-resistant cell colonies were allowed to form for 14 days before being fixed and stained with 0.1% crystal violet (Sigma-Aldrich Chemie NV, Zwijndrecht, The Netherlands, #C3886) in 20% methanol (VWR International B.V., Amsterdam, The Netherlands, #20847.320). Lentivirus genome copy titers were determined using the Lenti-X™ qRT-PCR Titration Kit (Takara Bio Europe SAS, Bio-Connect B.V., Huissen, The Netherlands, #631235) according to the manufacturer’s recommendations. 

Replication-defective adenovirus vector AdCMV-Luc [42] was generously provided by the Gene Therapy Center, University of Alabama at Birmingham, Birmingham, Alabama, USA. AdCMV-Luc is a first-generation E1/E3-deleted adenovirus vector (AdV) derived from human adenovirus serotype 5 (Ad5) carrying a CMV promoter-driven Firefly luciferase expression cassette in place of the deleted E1 region. Functional infectious unit (IU) titers were determined on E1-complementing 911 cells using the Adeno-X™ Rapid Titer Kit (Takara Bio Europe SAS, Bio-Connect B.V., Huissen, The Netherlands, #PT3651-2).

### 4.3. NSCLC-CAM Tumor Assay

Fertilized White Leghorn chicken eggs were purchased from Het Anker B.V. (Ochten, The Netherlands). Embryogenesis was considered to start when eggs were transferred to an incubator at a constant 37.5 °C and 55% humidity (EDD 0). From EDD 0 to EDD 3, the eggs were placed in a horizontal position on trays rotating 45° every 15 min. On EDD 3, the eggs were placed in a vertical position, and the trays were fixed. A small puncture hole was made in the top of the egg using a pointed tweezer and covered with adhesive tape. The eggs were incubated for another 3 days, during which the CAM under the puncture hole sank. On EDD 6, the top shell of the eggs was opened carefully using a pointed tweezer to create a window large enough to inspect the embryo and access the CAM. Fertilized chicken eggs exhibiting well-developed blood vessel networks on the CAM with viable embryos presenting strong heartbeats were selected for grafting NSCLC cells. Appendix A show examples of well-developed and poorly developed vessel networks, respectively. Next, a well-vascularized area in the center of the exposed CAM was cautiously damaged with clean paper tissue to create a small bleeding. NSCLC cells (in most experiments, 1 × 10^6^ cells per egg), suspended in 25 μL Matrigel^®^ Growth Factor Reduced Basement Membrane Matrix (Corning Life Sciences B.V., Amsterdam, The Netherlands, #354230) were applied inside the bleeding area. The eggs were closed again with adhesive tape and incubated until EDD 9 when embryo viability was confirmed, tumor take rate was documented, and, if applicable, eggs were randomized over treatment groups. Eggs with viable embryos and successfully grafted NSCLC tumors were monitored from EDD 9 to EDD 18 for embryo survival and tumor growth. To measure tumor volumes, tumors were observed under a ZEISS Stemi SV 6 stereo microscope with a mounted Axiocam 105 color camera (Zeiss Nederland, Breda, The Netherlands). When the entire tumor could be viewed through the shell window, its diameter was measured assuming an ellipse shape using ZEISS ZEN 2.6 software (accuracy of 0.1 mm). For tumors that could not be captured completely via the microscope, length and width dimensions were measured using a ruler (accuracy of 0.5 mm). The two measurement methods were calibrated by measuring the ruler scale using the software. Tumor dimensions measured using both methods were used to calculate tumor volumes according to the formula volume = (length × width^2^)/2. At EDDs indicated, after applying hypothermia for 20 min at −20 °C, tumors and/or embryo tissues were dissected, weighed, and fixed in 4% Formaldehyde (Sigma-Aldrich Chemie NV, Zwijndrecht, The Netherlands, #104003.100) for histology analysis (tumors) or snap-frozen in liquid nitrogen for qPCR analysis (all tissues); and embryos, from which only tumors were dissected, were sacrificed by freezing at −20 °C overnight. Experiments were never terminated later than EDD 18 to avoid any chance of hatching.

### 4.4. Histological Analysis of CAM Tumor Tissues

Tissues fixed in 4% formaldehyde were embedded in paraffin, cut using a microtome, mounted onto a slide, and stained with hematoxylin and eosin using standard procedures at the Pathology Department of the Amsterdam UMC. For specific staining, sections were deparaffinized in xylene (VWR International B.V., Amsterdam, The Netherlands, #28975.325) and rehydrated in graded ethanol (Supelco, VWR International B.V., Amsterdam, The Netherlands, #1.00983.2500) series. 

For LCA staining, first non-specific binding was blocked by incubating sections with Carbo-Free™ Blocking Solution (Vectorlabs, Bruschwig Chemie B.V., Amsterdam, The Netherlands, #SP-5040) for 30 min at room temperature. After removing the blocking solution from the sections and a single wash in PBS (Fresenius Kabi, Zeist, The Netherlands, #16PL9760), sections were incubated for 30 min in PBS supplemented with 2 μg/mL biotinylated LCA (Vectorlabs, Bruschwig Chemie B.V., Amsterdam, The Netherlands, #B-1045-5) and washed two times in PBST (PBS supplemented with 0.5% Tween 20 (Sigma-Aldrich Chemie NV, Zwijndrecht, The Netherlands, #8.22184.0500)) and once in PBS. Next, sections were incubated in VECTASTAIN^®^ ABC-AP alkaline phosphatase (Vectorlabs, Bruschwig Chemie B.V., Amsterdam, The Netherlands, #AK-5000) for 30 min at room temperature, followed by two washes in PBST and one in PBS. Finally, sections were incubated in ImmPACT Vector^®^ Red (Vectorlabs, Bruschwig Chemie B.V., Amsterdam, The Netherlands, #SK-5105) for 20 min and rinsed in tap water. 

For Ki-67 and APE1 staining, the mouse Specific HRP/DAB (ABC) Detection IHC Kit (Abcam, Cambridge, UK, #ab64259) was used. First, antigen retrieval was performed by boiling in Tris-EDTA buffer (10 mM Tris base (Sigma-Aldrich Chemie NV, Zwijndrecht, The Netherlands, #93352), 1 mM EDTA (AppliChem, VWR International B.V., Amsterdam, The Netherlands, #A1103), 0.05% Tween 20, pH 9.0) for 10 min using a microwave. Next, the Hydrogen Peroxide Block solution was applied for 10 min at room temperature, and after two washes in TBST with gentle agitation, Protein Block solution was applied for 10 min at room temperature. After a single wash in TBST, sections were incubated overnight at 4 °C with primary antibody against Ki-67 (Abcam, Cambridge, UK, #ab238020) or APE1 (Abcam, Cambridge, UK, #ab268072) diluted in 1% BSA (Sigma-Aldrich Chemie NV, Zwijndrecht, The Netherlands, #A9647) in PBS. After four washes in TBST with gentle agitation, sections were incubated in biotinylated goat anti-mouse antibody for 10 min at room temperature, washed 4 times in TBST, incubated in streptavidin-peroxidase conjugate for 10 min at room temperature, and rinsed 4 times in TBST. Finally, sections were stained with DAB Chromogen and DAB Substrate for 5 min, followed by rinsing 4 times in TBST. All specifically stained section slides were counterstained with Hematoxylin (Vectorlabs, Bruschwig Chemie B.V., Amsterdam, The Netherlands, #H-3404) and mounted using Eukitt^®^ Quick-hardening mounting medium (Sigma-Aldrich Chemie NV, Zwijndrecht, The Netherlands, #03989).

Slides were imaged using a Vectra Polaris Automated Quantitative Pathology Imaging System (PerkinElmer, Rotterdam, The Netherlands) or the SLIDEVIEW VS200 research slide scanner (Olympus Life Science, Hoofddorp, The Netherlands). Images were analyzed using QuPath software version 0.2.3 or 0.3.2 [43].

### 4.5. Bioluminescence Measurement via Live Imaging

All bioluminescence measurements were performed using an IVIS Lumina system (Xenogen, Xenogen, Alameda, CA, USA) (field of view D (12.5 cm); F/stop 1; medium binning) and analyzed using Living Image software (Xenogen, Alameda, CA, USA) (Living Image^®^ 4.5.2.18424). XenoLight D-Luciferin Potassium Salt Bioluminescent Substrate (PerkinElmer, Waltham, MA, USA, #1227991) was dissolved at 30 mg/mL in PBS and 30 microliters were added directly onto the CAM (not on the tumor tissue). Assuming an average egg weight of 50–55 g, this equaled a D-Luciferin dose of 16–18 mg/kg. Bioluminescence was measured after 20 min (exposure time: 15 s; maximum 4 eggs in one measurement), as pilot experiments (see Appendix A) had shown that, at this time, saturating substrate delivery to CAM tumors was reached, yielding stable luminescence signals. When measurements were performed on multiple days, fresh D-Luciferin substrate was administered each time.

### 4.6. Experimental Treatments of NSCLC-CAM Tumors

For chemotherapy experiments, eggs were weighed on the day of treatment. Pemetrexed (Sandoz, Almere, the Netherlands, #L01BA04) and/or Cisplatin (Accord, Harrow, UK, #15683354) was administered to the CAM surface in 100 μL 0.9% NaCl (Baxter B.V., Utrecht, The Netherlands, #TKF7124). In dose-finding embryo lethality experiments, chemotherapy was given on EDD 9 or EDD 11 at 1.5–20 mg/kg Pemetrexed and/or 0.1–3 mg/kg Cisplatin. NSCLC-CAM tumors were treated on EDD 11 with a single dose of 10 mg/kg Pemetrexed combined with 0.2 mg/kg Cisplatin. Tumors were observed under a microscope, and tumor sizes were measured from EDD 11 to EDD 18. Lentiviral vector LV-FLuc was diluted in 100 microliters of 0.9% NaCl and administered onto the CAM into the yolk sac or allantoic cavity or in 10 microliters of 0.9% NaCl and injected into the tumor tissue at the indicated dose on the indicated EDDs. Bioluminescence was measured 1 or 2 days after the last LV administration. Adenovirus vector AdCMV-Luc was administered at a single dose of 5 × 10^7^ IU on EDD 10, EDD 12, or EDD 17, either diluted in 100 microliter 0.9% NaCl and applied onto the CAM or in 5 microliter 0.9% NaCl and injected into the tumor. Bioluminescence was measured 1 day after virus addition, followed by tissue (tumor, exposed area of the CAM, lung, heart, liver, crop, kidney, and spleen) dissection from hypothermic embryos (eggs cooled at −20 °C for 20 min).

### 4.7. Quantification of Adenovirus Copy Numbers via qPCR

Dissected and weighed tumors and embryo tissues were fresh-frozen in liquid nitrogen before their homogenization using a Precellys Evolution Tissue Homogenizer (Bertin Technologies, VWR International B.V., Amsterdam, The Netherlands, P000062-PEVO0-A) and 2 mL Hard Tissue homogenizing CK28 tubes (Bertin Technologies, VWR International B.V., Amsterdam, The Netherlands, P000911-LYSK0-A). DNA was isolated from tissue homogenates using the QIAamp DNA Blood Mini Kit (Qiagen Benelux B.V., Venlo, The Netherlands, #51106) according to the manufacturer’s recommendation, with the proteinase K incubation step prolonged to 1 h to ensure complete digestion of tissue samples. DNA was eluted in 50 µL elution buffer, and 5 µL was used for quantification of adenovirus copy number via qPCR using 5× HOT FIREPol^®^ EvaGreen^®^ qPCR Mix Plus (no ROX) (Solis BioDyne, Bio-Connect, Huissen, The Netherlands, #08-25-00008) and primers detecting the adenovirus serotype 5 packaging domain (FWD: 5′-GGA AGT GAC AAT TTT CGC GC-3′; REV: 5′-CCC GCG GCC CTA GAC AAA TAT-3′). qPCR reactions were run on a LightCycler^®^ 480 System (Roche Diagnostics Nederland B.V., Almere, The Netherlands). Adenovirus copy numbers are presented as mean 2^−Cp^ values of technical duplicates.

### 4.8. Statistical Analysis

All data were analyzed using GraphPad Prism software version 9.3.1 (471). The statistical tests applied are given with the description of the results.

Power analysis to determine sample size in treatment experiments was conducted on the observed growth of untreated tumors, assuming equal variances, using the formula N = 2 × [(Z_α_+Z_1-ß_) × σ/δ]^2^.

In the chemotherapy experiments, normal distribution of relative tumor volumes was evaluated using the D’Agostino-Pearson test (*p* > 0.05, all passed normality test); the chemotherapy effects on A549 tumor growth were tested using the unpaired-sample *t*-test (two groups were compared, independent data, two-tailed, F test to compare variances: *p* = 0.49); the chemotherapy effects on H460 tumor growth was tested using the unpaired-samples *t*-test with Welch’s correction for unequal variances (two groups were compared, independent data, two-tailed, F test to compare variances: *p* < 0.01).

The lognormal distribution of luminescence intensities on LV-Fluc transduced NSCLC-CAM tumors at 24 and 48 hpi was evaluated using the D’Agostino-Pearson test (*p* > 0.05, passed lognormality test), and the comparison of luminescence measurements at the two time points was carried out using the paired *t*-test (two-tailed).

The mean rank of bioluminescence signal intensity of NSCLC-CAM tumors transduced with LV-Fluc in different dose groups was compared using the Kruskal–Wallis test (6 dose groups, independent data). The mean rank of bioluminescence signal intensity of tumor tissues transduced with AdCMV-Luc administered intratumoral or via the CAM on the same EDD was compared using the Mann–Whitney test (2 administration routes, independent data), and the mean rank of tumor bioluminescence signal intensity between different EDDs using the same administration route was compared using the Kruskall-Wallis test (3 different EDDs, independent data).

The mean rank of AdCMV-Luc copy numbers in different embryo tissues on the same EDDs, in the same tissue via different administration routes on the same EDDs, in the same tissue via the same administration route in different models (with A549 or H460 tumor, or without tumor) on the same EDDs, or in the same tissues via the same administration routes on different EDDs was compared using the Kruskal–Wallis test (more than 2 groups, independent data).

## 5. Conclusions

Our study shows that NSCLC-CAM tumors can be grown reproducibly and that established NSCLC-CAM tumors provide workable in vivo assays to rapidly explore novel anti-tumor therapies. Chemical molecules and viral vectors are delivered to NSCLC-CAM tumors and embryo tissues via systemic administration. Efficient gene transfer with recombinant adenovirus vectors makes the NSCLC-CAM tumor model attractive for use in gene therapy studies. The relative simplicity and low cost of the model enable the testing of multiple conditions in parallel. NSCLC-CAM tumors are, thus, very useful tools for preclinical lung cancer research.

## Figures and Tables

**Figure 1 ijms-24-15425-f001:**
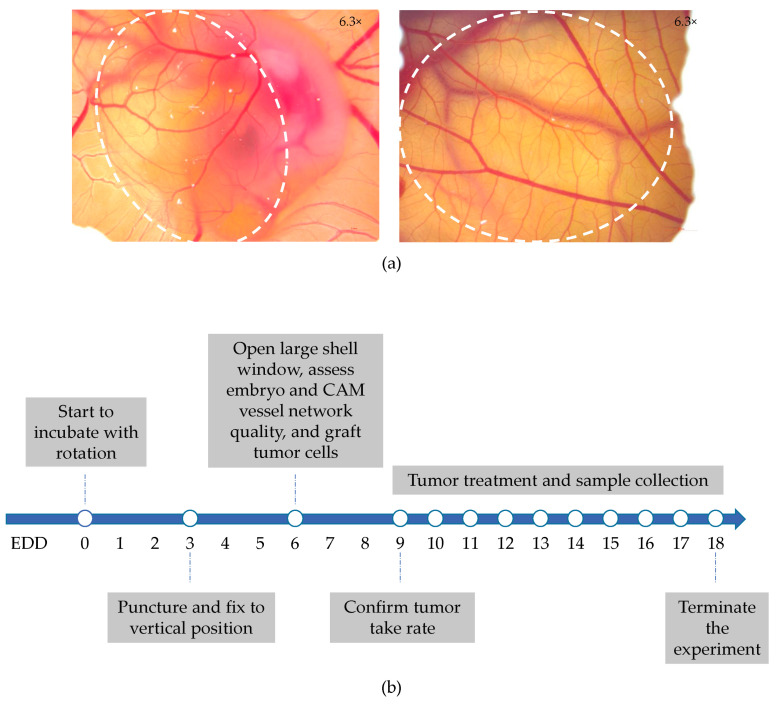
The CAM tumor model. (**a**) Microscopic images of the chick embryo and extra-embryonic vasculature on EDD 6 (left) and EDD 10 (right). The position of the CAM protruding from the embryo is encircled. Images were taken through the shell window at 6.3× magnification. (**b**) Typical time schedule for a CAM tumor experiment.

**Figure 2 ijms-24-15425-f002:**
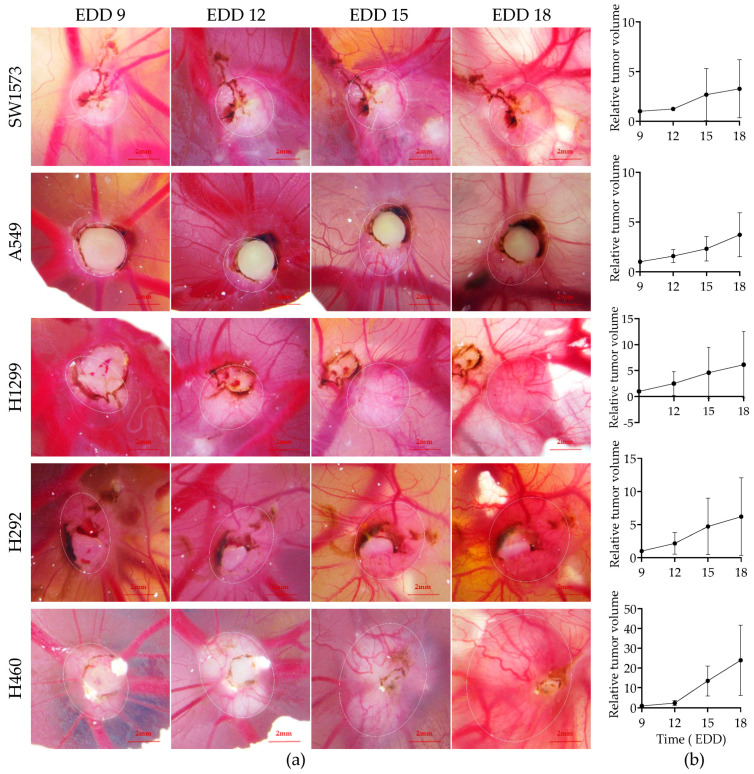
NSCLC-CAM tumor growth. (**a**) microscopic images of representative tumors established from 5 NSCLC cell lines growing on the CAM taken on the EDDs indicated. The tumor dimensions, defined with the ellipse drawn in the ZEISS ZEN 2.6 software, are indicated. (**b**) Growth rates of the CAM-NSCLC tumors from EDD 9 to EDD 18 (n = 7–13 tumors per group). The graphs show average (±SD) tumor volumes normalized via the starting volume at EDD 9. Tumors are ranked from the slowest (top) to fastest (bottom) growing NSCLC cell line.

**Figure 3 ijms-24-15425-f003:**
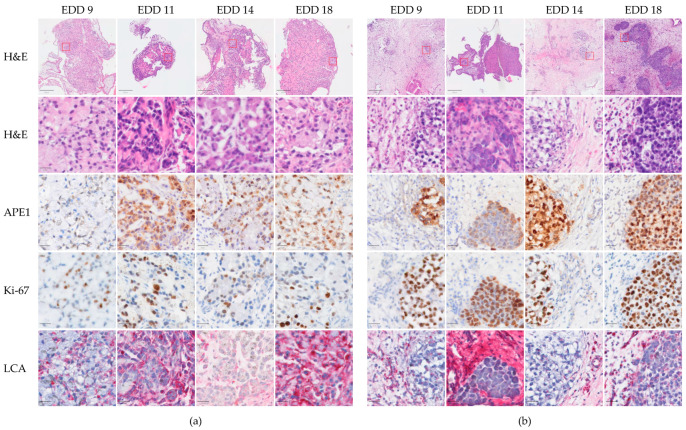
(Immuno)histological analysis of A549 (**a**) and H460 (**b**) NSCLC-CAM tumors. NSCLC cells were inoculated on the CAM in Matrigel on EDD 6 and allowed to form tumors. Established tumors on EDD 9 were followed until EDD 18. Representative examples of tumors dissected at the indicated EDD are shown. For their analysis, 4 μm sections from the same paraffin-embedded tumor were stained with hematoxylin-eosin or immunostained for APE1, Ki-67, or LCA as indicated. Positive APE1 or Ki-67 staining is indicated in brown, LCA in red, and hematoxylin counterstain in blue. Scale bars in the top rows are 250 μm; in all other rows, scale bars are 20 μm. The regions selected for magnification are depicted in the top row images.

**Figure 4 ijms-24-15425-f004:**
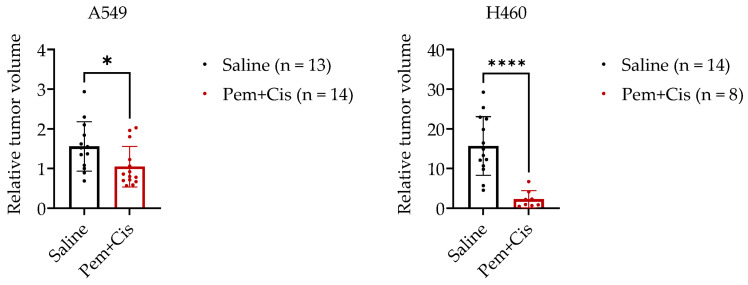
Treatment of NSCLC-CAM tumors with systemic chemotherapy. A549-CAM tumors and H460-CAM tumors were treated with 10 mg/kg Pemetrexed and 0.2 mg/kg Cisplatin administered onto the CAM on EDD 11. Tumor volumes were determined at EDD 11 and EDD 17. The graphs depict the relative tumor growth from EDD 11 to EDD 17. Results of individual tumors are shown as dots and mean results with SD as bars with whiskers. *, *p* < 0.05; ****, *p* < 0.0001 (unpaired-samples *t*-test).

**Figure 5 ijms-24-15425-f005:**
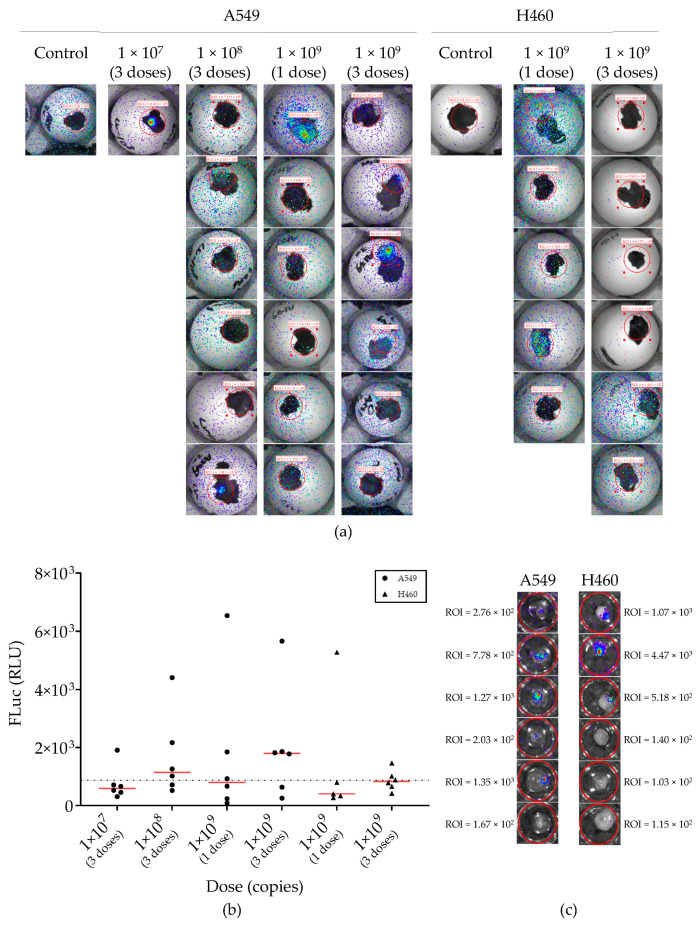
Lentiviral vector delivery to NSCLC-CAM tumors. (**a**) IVIS images of eggs bearing A549/H460-CAM tumors that were treated with different doses of LV-FLuc and negative controls. LV-FLuc was administered on EDD 9, EDD 11, and EDD 13 (3 doses) to A549-CAM- or H460-CAM-bearing eggs at the indicated dose (1 × 10^7^, 1 × 10^8^, or 1 × 10^9^) or at 1 × 10^9^ copies only once on EDD 9; luminescence was measured 48h after the last dose. (n = 6 eggs per group; eggs from the 1 × 10^7^ group that did not emit luminescence above background are not shown. (**b**) Quantification of the bioluminescence. Region of interest (ROI) was set at 30 pixels in diameter. Graph shows individual luminescence measurements with medians. The dashed line indicates the maximum background measured in these experiments (870 RLU). (**c**) 1 × 10^9^ copies LV-FLuc were administered daily from EDD 9 to EDD 13 (5 doses) to A549-CAM- or H460-CAM-bearing eggs. Tumors were dissected and imaged 48 h after the last dose.

**Figure 6 ijms-24-15425-f006:**
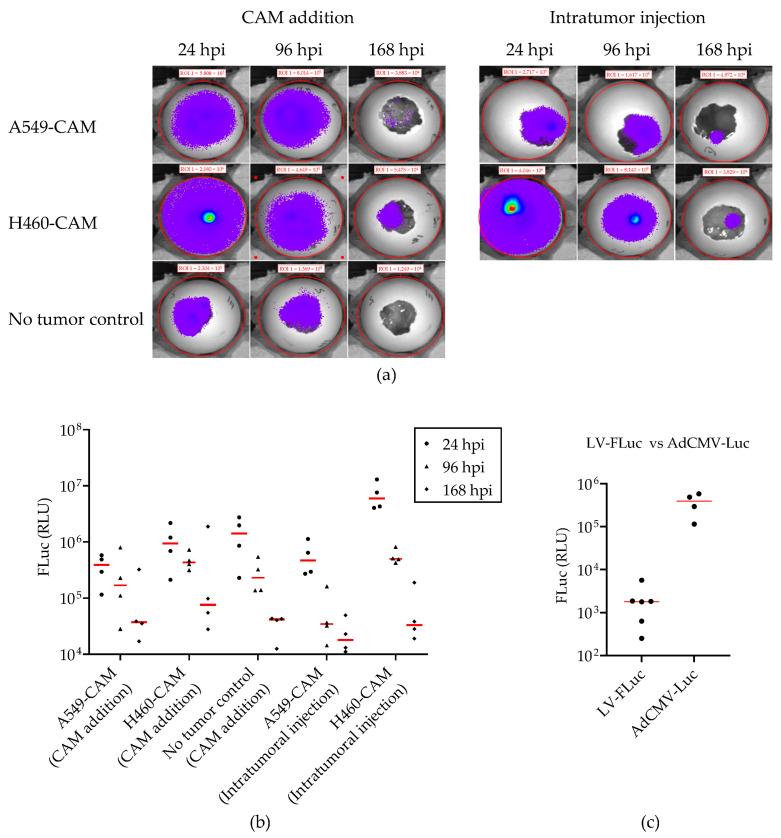
AdCMV-Luc transduction efficiency in NSCLC-CAM tumor models. (**a**) AdCMV-Luc (5 × 10^7^ IU) was applied to fertilized chicken eggs with or without established A549 or H460 tumors via CAM addition or intratumoral injection on EDD 10. Bioluminescence was measured at the indicated time points after AdCMV-Luc administration. Representative images with IVIS measurement values of individual eggs are shown. (**b**) Quantification of bioluminescence in the different treatment groups (ROI: 100 pixels diameter; n = 4 eggs per group). (**c**) Comparison of luminescence measured in A549-CAM tumors treated by administering 5 × 10^7^ IU AdCMV-Luc to the CAM on EDD 10 (data from Figure 6b) versus three times CAM delivery of 1 × 10^9^ copies LV-FLuc on EDD 9, EDD 11, and EDD 13 (highest median luminescence reached with LV-FLuc; data from Figure 5).

**Figure 7 ijms-24-15425-f007:**
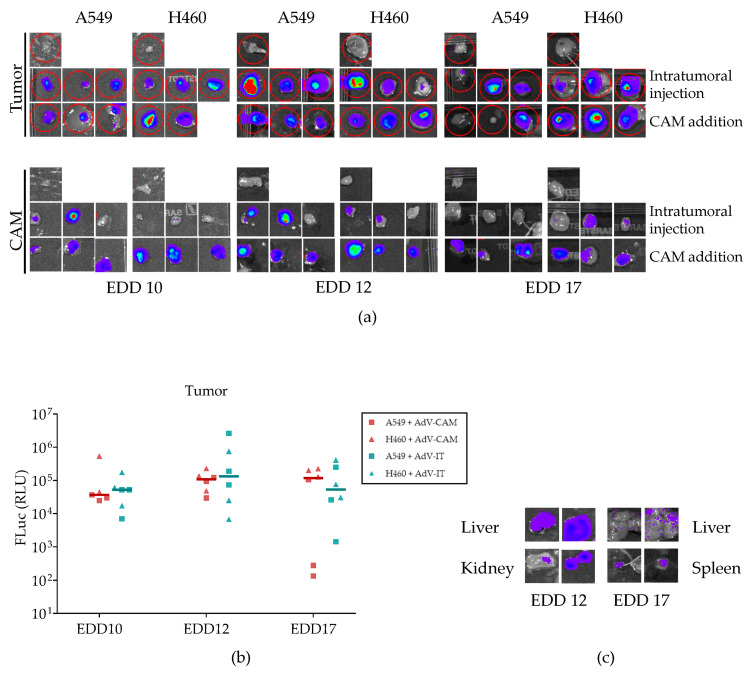
Bioluminescence imaging of dissected tissues. (**a**) IVIS images of all dissected tumors and CAM tissues. (**b**) Bioluminescence signal intensity of tumor tissues transduced with AdCMV-Luc administered intratumoral (IT) or via the CAM at the indicated EDDs and imaged the next day. Differences between the two administration methods on the same day (Mann–Whitney test) or between different days using the same administration route (Kruskal–Wallis test) were not significant. (**c**) IVIS images showing detectable bioluminescence in the liver and kidney on EDD 12 (left) and liver and spleen on EDD 17 (right). N = 3 eggs per treatment group (one tumor sample lost in the EDD10 H460-CAM group); with a single negative control egg for each NSCLC cell line on every analysis day.

**Figure 8 ijms-24-15425-f008:**
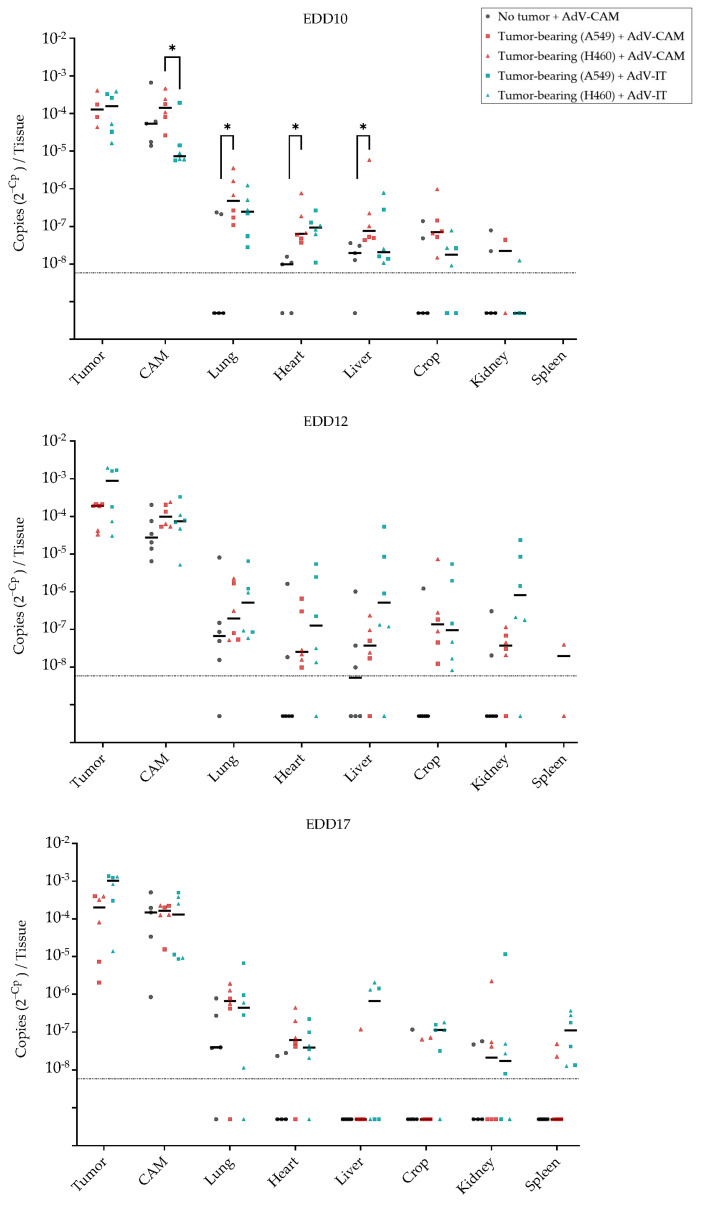
Biodistribution of AdCMV-Luc administered at 3 different time points to chicken embryos. AdCMV-Luc was administered to fertilized eggs with or without A549 or H460-CAM tumors either via application onto the CAM (AdV-CAM) or intratumoral injection (AdV-IT) on EDD 10, EDD 12, or EDD 17, as indicated. One day later, after measuring luciferase expression (see Figure 7), different tissues (tumor, CAM, lung, heart, liver, crop, kidney, spleen) were dissected and analyzed via qPCR. N = 3 eggs per treatment group (one tumor sample lost in the EDD10 H460-CAM and A459-CAM groups); with 5–6 negative control eggs on every analysis day. Spleens could not be dissected at EDD 10 and from only few embryos on EDD 12. AdCMV-Luc copy numbers are presented as 2^–Cp^ values. For each treatment group, individual data points and medians (lines) are shown. The dashed line indicates the limit of detection. Negative qPCR results are depicted at an arbitrary position below this line. *, *p* < 0.05 (Kruskal–Wallis test).

**Table 1 ijms-24-15425-t001:** Characteristics of human NSCLC-CAM tumor models.

Cell Line	Tumor Take Rate on EDD 9	Percent Measurable Tumors on EDD 9	Embryo Survival Until EDD 18 *	Mean Relative Tumor Volume Increase from EDD 9–18 (SD)	Sample Size Needed Derived from Power Analysis ^#^
SW1573	95% (40/42)	55% (22/40)	95% (21/22)	3.3 (2.9)	28
A549	81% (34/42)	65% (22/34)	91% (20/22)	3.7 (2.2)	13
H1299	91% (21/23)	57% (12/21)	100% (12/12)	6.1 (6.4)	37
H292	95% (20/21)	50% (10/20)	100% (10/10)	6.2 (5.9)	31
H460	90% (38/42)	45% (17/38)	88% (15/17)	23.9 (17.7)	19

* Viable embryos on EDD 18 that had measurable tumors on EDD 9. ^#^ Power analysis was conducted on the basis of the observed SD/mean (σ) and the following predefined parameters: single-sided test assuming equal variances, α = 0.05; β = 0.2; δ = 0.6.

## Data Availability

The data presented in this study are available in the article and Appendix A.

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
