# Peer review of "Human Non-Small Cell Lung Cancer-Chicken Embryo Chorioallantoic Membrane Tumor Models for Experimental Cancer Treatments"

_ijms, 2023, doi:10.3390/ijms242015425_

Round 1

Reviewer 1 Report (New Reviewer)

The work is written in an understandable and orderly manner. The concept of the work is well constructed in terms of content. The materials and methods as well as the results are comprehensively described. The results are confirmed by photographic documentation. The study is very valuable as it allows limiting research on experimental animals before clinical trials. The work may be accepted in its current form.

Author Response

We are happy with the positive evaluation of our work by the reviewer.

Reviewer 2 Report (New Reviewer)

This is a very interesting work, well designed with good experimental setting and a professional writing. Proposing the use of these in ovo models to replace animals is a great direction for obvious ethical reasons. These models have been extensive proposed in the literature including human lung cancers. Therefore, reading the manuscript raised one main question. What is the novelty of this work when compared to the established literature for this field?  

While this model applies for toxicity study, the authors must include a deep discussion for the limitations. In the Introduction section (page 2), the authors claim that these models “are generally considered most appropriate for the validation of new treatments before clinical translation". Therefore, how these models can test many other critical features of a drug beside efficacy/toxicity like pharmacokinetics/pharmacodynamics, metabolism, drug interactions etc.? Are these models useful for drugs, which are effective only as active metabolite? Discussion related to the used of other treatments such radiotherapy/immunotehrapy must be included/extended.

Finally, I see that a lot of important publications in the field are not cited.

Author Response

This is a very interesting work, well designed with good experimental setting and a professional writing. Proposing the use of these in ovo models to replace animals is a great direction for obvious ethical reasons.

We are pleased with the positive opinion of the reviewer on our manuscript.

These models have been extensive proposed in the literature including human lung cancers. Therefore, reading the manuscript raised one main question. What is the novelty of this work when compared to the established literature for this field?

Triggered by the remark made by the reviewer, we searched the literature for other studies reporting NSCLC-CAM tumor models and their use to test treatments. We identified several relevant papers published in recent years. The comment by the reviewer is thus correct. In our revised manuscript, we include a new paragraph in the discussion section on page 14 (highlighted) in which we discuss these studies (new cited references #30-33). In line with the established use of the CAM model to study neovascularization, two of these studies focused on antiangiogenic effects of tested compounds. The third investigation used siRNA-transfected NSCLC cells grafted on the CAM; and in the fourth study a compound was applied topically onto the tumors. Our work extends these previous studies (i) by providing detailed description of the methods used to allow readers to establish and use these models; (ii) by quantifying tumor take and growth of five different cell lines; (iii) by treating established tumors systemically rather than grafting treated cells or treating grafted cells during the first 2 days when tumors are formed or by topical application; and, most importantly, (iv) by evaluating the models for their utility in gene delivery experiments.

We also identified a publication in which metastasis of NSCLC cells in the chicken embryo was investigated. Although the approach and goal are different from ours, we consider also this interesting work relevant to be mentioned (cited in the same paragraph as new reference #34).

While this model applies for toxicity study, the authors must include a deep discussion for the limitations. In the Introduction section (page 2), the authors claim that these models “are generally considered most appropriate for the validation of new treatments before clinical translation". Therefore, how these models can test many other critical features of a drug beside efficacy/toxicity like pharmacokinetics/pharmacodynamics, metabolism, drug interactions etc.? Are these models useful for drugs, which are effective only as active metabolite?

There is an apparent misunderstanding here. The citation is taken from the introduction section, where we wrote “In contrast, animal models for cancer allow the investigation of novel treatments in the context of a complete organism. A large variety of rodent, mainly mouse, models are used for this purpose [10]. Researchers can either establish syngeneic tumors in immune-competent animals or xenograft tumors in immune-deficient animals. While also these models have their limitations, they are generally considered most appropriate for the validation of new treatments before clinical translation.” We were thus referring to animal models, in particular mouse models; not to CAM tumor models in fertilized chicken eggs. We then continued, stating “However, considering their complexity and high costs and the strong ethical wish to reduce the use of experimental animals, animal models are less preferred to test a multitude of agents, concentrations, or treatment schedules. For this, alternative in vivo tumor models that can bridge initial cell culture experiments to validation in animal models are very useful.” This brought us to introducing the NSCLC-CAM tumor models. We present them as an intermediate step between in vitro cultures and animal models, reducing the number of animals needed in the final validation experiments. We do not wish to suggest in any way that CAM tumor models will replace animal models entirely. The reviewer is correct in arguing that e.g. PD/PK studies in drug development may still require animal experiments. The reviewer is also correct in suggesting that an embryonic model could perhaps not always be suitable to investigate drugs that need to be converted into an active metabolite. It is foreseeable that the developing embryonic liver might not yet fully support drug activation.

Discussion related to the used of other treatments such radiotherapy/immunotehrapy must be included/extended.

Immunotherapy is indeed an important emerging option in NSCLC. In the last paragraph of our discussion, we shortly discussed the opportunity to also perform immunotherapy experiments in NSCLC-CAM tumor models during the last few days of embryo development, when a physiologically reactive immune system develops. Following the request of the reviewer, we have extended this discussion with a realistic assessment of the probable limits of such investigations (addition highlighted in the revised manuscript).

Radiotherapy treatments are likely to be also possible in the presented NSCLC-CAM tumor models, as this was already done. In addition to previously cited reference #20, in which radiotherapy and anti-angiogenic treatment schedules were investigated in CAM tumor models for colorectal cancer and esophageal adenocarcinoma, we now also found a report in which NSCLC-CAM tumors growing ex ovo were irradiated and treated with an anti-angiogenic antibody. In our revised manuscript, we mention this shortly in the discussion section (highlighted), but do not elaborate on it, as we have not performed any radiotherapy experiments.

Finally, I see that a lot of important publications in the field are not cited.

This could be true; and we apologize for not citing contributions by others that deserve attention. In our opinion, however, we do provide essential references to support the statements made in our manuscript. If, however, we have missed a publication that is crucial to be cited, we appreciate it if we could be specifically pointed at this work. We will then consider to include it.

Reviewer 3 Report (New Reviewer)

While the approach to establishing human NSCLC xenograft tumor assays on the CAM of chicken embryos is innovative and could potentially advance our understanding of NSCLC treatment, the manuscript has several limitations and areas that require thorough revision and clarification.

1.     The use of chicken embryos as a host for human tumor cell xenografts raises concerns about the relevance and transferability of the findings to human physiology and disease contexts. Clarification is needed on why this model was chosen over more established models and how it replicates the human NSCLC environment. Furthermore, the experimental design lacks adequate controls, both positive and negative, to validate the effects observed. This includes controls for nonspecific effects and potential toxicities related to the treatments administered.

2.     The study lacks appropriate control groups and standardized conditions for comparison, crucial for validating the experimental results. Standardization of conditions like temperature, humidity, and inoculation volume would enhance the credibility of the findings. In additions, the rationale behind the selection of the A549 and H460 cell lines needs to be clearly articulated. Additionally, the use of only two cell lines limits the generalizability of the findings. It would be beneficial to further validate your NSCLC-CAM tumor models with more NSCLC cell lines to ensure that these findings are generalizable across multiple NSCLC subtypes.

3.     The paper demonstrates the utility of adenoviral vectors over lentiviral vectors in transducing NSCLC tumors, but fails to address the specificity of adenovirus transduction in cancer cells versus healthy cells, which is critical for assessing the therapeutic window.

4.     Clearer elucidation on the dosage of chemotherapy and the rationale behind the chosen dosage is needed. Details on the administration method, frequency, and duration are also lacking. Detailed protocols for the administration of Pemetrexed and Cisplatin, including dose rationalization, are lacking and are critical for interpreting the results correctly. The paper needs a clearer rationale and more detailed information about the choice of drug dosages and administration routes. How they align with clinical scenarios is also not addressed sufficiently.

5.     The manuscript would benefit from a more in-depth exploration and elucidation of the mechanisms underlying the observed effects, potentially through the incorporation of molecular studies investigating the interaction between the vectors and NSCLC cells.

6.     The study does not provide a comparative analysis with other established preclinical models to demonstrate the added value or superiority of the NSCLC-CAM model in assessing new cancer therapies. A more extensive comparison with existing preclinical models for NSCLC is needed, along with a clear justification for the advantages of the proposed CAM model.

7.     The study offers minimal discussion on the clinical relevance and applicability of its findings. A comprehensive understanding of how this preclinical model could be translated to human applications is pivotal for its acceptance in the scientific community. There is a notable absence of discussion on how the findings from this model can be translated to clinical settings, and how they align or contrast with existing knowledge and therapeutic strategies for NSCLC.

8.     A critical evaluation of the safety and potential toxicities of the systemic treatments and vector deliveries in the model is missing and needs to be addressed.

9.     The study lacks detailed information on adherence to ethical standards and guidelines regarding the use of animal models. It is crucial to ensure that all experimental procedures were conducted following relevant ethical norms. It would be pertinent to discuss any ethical considerations, particularly regarding the use of embryos, ensuring that animal welfare guidelines were followed.

10.It is crucial to acknowledge the potential limitations and shortcomings of your approach, such as the differences between chicken embryonic and human environments and any potential biases in your study. A more detailed and thoughtful discussion on the limitations of the study, including potential biases and their impact on the results, is essential.

Given the multiple areas of concern, substantial revisions and potentially additional experiments are required to address the shortcomings and validate the findings adequately. The paper must address these concerns with due diligence to elevate its scientific validity and translational potential.

While the approach to establishing human NSCLC xenograft tumor assays on the CAM of chicken embryos is innovative and could potentially advance our understanding of NSCLC treatment, the manuscript has several limitations and areas that require thorough revision and clarification.

1.     The use of chicken embryos as a host for human tumor cell xenografts raises concerns about the relevance and transferability of the findings to human physiology and disease contexts. Clarification is needed on why this model was chosen over more established models and how it replicates the human NSCLC environment. Furthermore, the experimental design lacks adequate controls, both positive and negative, to validate the effects observed. This includes controls for nonspecific effects and potential toxicities related to the treatments administered.

2.     The study lacks appropriate control groups and standardized conditions for comparison, crucial for validating the experimental results. Standardization of conditions like temperature, humidity, and inoculation volume would enhance the credibility of the findings. In additions, the rationale behind the selection of the A549 and H460 cell lines needs to be clearly articulated. Additionally, the use of only two cell lines limits the generalizability of the findings. It would be beneficial to further validate your NSCLC-CAM tumor models with more NSCLC cell lines to ensure that these findings are generalizable across multiple NSCLC subtypes.

3.     The paper demonstrates the utility of adenoviral vectors over lentiviral vectors in transducing NSCLC tumors, but fails to address the specificity of adenovirus transduction in cancer cells versus healthy cells, which is critical for assessing the therapeutic window.

4.     Clearer elucidation on the dosage of chemotherapy and the rationale behind the chosen dosage is needed. Details on the administration method, frequency, and duration are also lacking. Detailed protocols for the administration of Pemetrexed and Cisplatin, including dose rationalization, are lacking and are critical for interpreting the results correctly. The paper needs a clearer rationale and more detailed information about the choice of drug dosages and administration routes. How they align with clinical scenarios is also not addressed sufficiently.

5.     The manuscript would benefit from a more in-depth exploration and elucidation of the mechanisms underlying the observed effects, potentially through the incorporation of molecular studies investigating the interaction between the vectors and NSCLC cells.

6.     The study does not provide a comparative analysis with other established preclinical models to demonstrate the added value or superiority of the NSCLC-CAM model in assessing new cancer therapies. A more extensive comparison with existing preclinical models for NSCLC is needed, along with a clear justification for the advantages of the proposed CAM model.

7.     The study offers minimal discussion on the clinical relevance and applicability of its findings. A comprehensive understanding of how this preclinical model could be translated to human applications is pivotal for its acceptance in the scientific community. There is a notable absence of discussion on how the findings from this model can be translated to clinical settings, and how they align or contrast with existing knowledge and therapeutic strategies for NSCLC.

8.     A critical evaluation of the safety and potential toxicities of the systemic treatments and vector deliveries in the model is missing and needs to be addressed.

9.     The study lacks detailed information on adherence to ethical standards and guidelines regarding the use of animal models. It is crucial to ensure that all experimental procedures were conducted following relevant ethical norms. It would be pertinent to discuss any ethical considerations, particularly regarding the use of embryos, ensuring that animal welfare guidelines were followed.

10.It is crucial to acknowledge the potential limitations and shortcomings of your approach, such as the differences between chicken embryonic and human environments and any potential biases in your study. A more detailed and thoughtful discussion on the limitations of the study, including potential biases and their impact on the results, is essential.

Given the multiple areas of concern, substantial revisions and potentially additional experiments are required to address the shortcomings and validate the findings adequately. The paper must address these concerns with due diligence to elevate its scientific validity and translational potential.

Author Response

While the approach to establishing human NSCLC xenograft tumor assays on the CAM of chicken embryos is innovative and could potentially advance our understanding of NSCLC treatment, the manuscript has several limitations and areas that require thorough revision and clarification.

We are pleased that the reviewer recognizes the novelty of our work. Below, we respond to the 10 issues raised by the reviewer.

  1. The use of chicken embryos as a host for human tumor cell xenografts raises concerns about the relevance and transferability of the findings to human physiology and disease contexts. Clarification is needed on why this model was chosen over more established models and how it replicates the human NSCLC environment.

As presented in the introduction of our manuscript, we do not choose NSCLC-CAM tumor models over xenograft models in other species such as mice. In our view, NSCLC-CAM tumor models provide a very useful intermediate step between in vitro cultures and animal models, reducing the number of animals needed in the final validation experiments. This brings forth practical as well as ethical advantages (see our answer to point 6). In terms of relevance of results obtained in NSCLC-CAM tumor models for the human disease context, this is very similar to the relevance of other xenograft tumor models. One should always be careful in extrapolating observations made on human cancers growing in tissue of a different species. Interactions in the tumor microenvironment might not be the same as in the human body. Experimental treatments of NSCLC-CAM tumors can be investigated during the mid and late stages of embryo development. In the midstage, the models are best compared to xenograft models in immune deficient animals such as nude mice; in the late stage the models provide the benefit of an immune competent context.

Furthermore, the experimental design lacks adequate controls, both positive and negative, to validate the effects observed. This includes controls for nonspecific effects and potential toxicities related to the treatments administered.

This remark surprises us. All experiments included the proper controls to interpret the results. These were obviously negative controls (no NSCLC cell inoculation, no chemotherapy, no viral vector administration). We did not include any positive controls. There was no luciferase expression vector control already known to efficiently transduce established NSCLC-CAM tumors (if there were, we would not have tested LV and AdV); and we do not consider it useful nor ethical to treat embryos with a highly toxic irrelevant compound as positive control for toxicity.

  1. The study lacks appropriate control groups and standardized conditions for comparison, crucial for validating the experimental results. Standardization of conditions like temperature, humidity, and inoculation volume would enhance the credibility of the findings. In additions, the rationale behind the selection of the A549 and H460 cell lines needs to be clearly articulated. Additionally, the use of only two cell lines limits the generalizability of the findings. It would be beneficial to further validate your NSCLC-CAM tumor models with more NSCLC cell lines to ensure that these findings are generalizable across multiple NSCLC subtypes.

As said, controls were included everywhere. We thus disagree with the reviewer on this point. Details about temperature, humidity, volumes, etcetera are all given in the materials and methods section. The rationale for choosing the A549 and H460 models over the other three NSCLC-CAM models for testing experimental treatments is clearly mentioned in the results section on page 4 in our manuscript: “A549 lung adenocarcinoma and H460 large cell lung cancer CAM tumor models, representing one of the two slowest growing NSCLC-CAM tumor models and the fastest growing NSCLC-CAM tumor model, respectively, exhibited the smallest variation in growth rate, hence requiring the smallest group sizes. Therefore, A549-CAM and H460-CAM models are preferred choices when performing tumor growth inhibition experiments with multiple treatment groups.” and on page 14 in the discussion section “All five human NSCLC cell lines included in our studies rapidly formed solid tumors on the CAM and their growth could be followed until the termination of the experiment. Of these, A549-CAM and H460-CAM tumors grew most reproducibly, with an acceptable variation requiring the smallest group sizes to detect significant effects of anti-cancer treatments. Therefore, these two NSCLC-CAM models were chosen for further studies.” It is not entirely clear which generalizable findings the reviewer is hinting at. The findings that tumors derived from human NSCLC cell lines can be engrafted and grown reproducibly on the CAM of fertilized chicken eggs; and that >90% tumor-bearing embryos remain vital over the entire course of the experiment, can probably be generalized, as these observations were made with five different cell lines. The findings that NSCLC-CAM tumor growth can be inhibited by systemic chemotherapy and that genes can be delivered to NSCLC-CAM tumors and expressed in these tumors using adenoviral vectors much more efficiently than using lentiviral vectors were indeed only obtained in studies on two models, but were at least reproducible in these models. Nevertheless, we observed differences between the two models. As we have described in our manuscript, A549-CAM and H460-CAM tumors exhibit distinct histology and very different growth rates. It was thus not unexpected that we measured more significant growth inhibition by chemotherapy treatment for the faster growing H460-CAM tumors. We also observed some differences (although not statistically significant) in gene delivery efficiencies between the two models. Therefore, these observations can probably be generalized qualitatively, but not necessarily quantitatively. Thus, if for example gene delivery is considered in a NSCLC-CAM tumor model produced with another NSCLC cell line, on the basis of our findings we dare to advise that in these experiments an adenovirus vector is a better choice than a lentiviral vector, but we cannot predict which gene delivery efficiency will be achieved.

  1. The paper demonstrates the utility of adenoviral vectors over lentiviral vectors in transducing NSCLC tumors, but fails to address the specificity of adenovirus transduction in cancer cells versus healthy cells, which is critical for assessing the therapeutic window.

On this point, we have a different opinion. In our view, for a proper evaluation of the efficacy and safety of therapeutic gene delivery it is better to deliver the genetic material (e.g., a transgene or silencing molecule), to cancer cells as well as healthy cells. This will allow assessing anticancer efficacy and toxicity. We mentioned this in the last paragraph of the discussion section, stating that the use of AdV “will maximize gene expression in CAM tumors, providing the best chance to assess anti-cancer treatment efficacy; and allow analysis of the safety of therapeutic gene delivery by studying toxicity to non-malignant tissues.” Please note that in such preclinical studies the viral vector is only used as gene delivery tool. The introduced genetic material (e.g., an shRNA silencing a therapeutic target or a gene expressing an apoptosis-inducing protein or a prodrug converting enzyme) provides the anti-cancer effect. In addition, an approach that is increasingly being tested in preclinical and early clinical studies, is the use of oncolytic adenoviruses. There, the therapeutic window is usually not determined by cancer-specific transduction, but by cancer-selective replication. Our observation of very efficient systemic adenovirus delivery to NSCLC-CAM tumors suggest that these models could also be used to test oncolytic adenoviruses in vivo. Notably, this could also include experiments to test capsid-engineered viruses with modified tissue tropism.

  1. Clearer elucidation on the dosage of chemotherapy and the rationale behind the chosen dosage is needed. Details on the administration method, frequency, and duration are also lacking. Detailed protocols for the administration of Pemetrexed and Cisplatin, including dose rationalization, are lacking and are critical for interpreting the results correctly. The paper needs a clearer rationale and more detailed information about the choice of drug dosages and administration routes. How they align with clinical scenarios is also not addressed sufficiently.

We are surprised by this comment, because all details asked for are given in our manuscript. The administration method (single application to the CAM surface in 100 μl 0.9% NaCl) is described in Materials and Methods section 4.6. The rationale for the drug doses used (10 mg/kg Pemetrexed combined with 0.2 mg/kg Cisplatin) and the day of administration (EDD11), based on the pilot safety experiments testing 1.5-20 mg/kg Pemetrexed and/or 0.1-3 mg/kg Cisplatin given on EDD9 or EDD11 shown in Figure S2, is given on page 6-7 of our manuscript. We chose the highest dose combination that did not kill any embryo in the dose-finding experiments. The text in our manuscript reads: “Since this treatment had not been tested in a CAM tumor model before, we first assessed a safe dose regimen in fertilized chicken eggs. Cisplatin and/or Pemetrexed were administered to the CAM on EDD 9 or 11 at various concentrations, and embryo vitality was monitored from EDD 9 to EDD 18 (Figure S2). The main criteria for vitality were the heartbeat of the embryo, clear and light red blood vessels (indicating circulation of fresh oxygenated blood), and the presence of small vessels along the shell. Chemotherapy given on EDD 9 was highly toxic, except for low-dose treatment with Cisplatin only. In contrast, when chemotherapy was delayed until EDD 11, 100% embryo survival was observed at combination treatment concentrations up to 10 mg/kg Pemetrexed with 0.2 mg/kg Cisplatin. Therefore, this dose combination was used to investigate tumor growth inhibition in A549-CAM and H460-CAM models.” Furthermore, as mentioned in the discussion section on page 14 of our manuscript, the doses we used in the CAM tumor models were considerably lower than doses that are generally used in mouse models.

  1. The manuscript would benefit from a more in-depth exploration and elucidation of the mechanisms underlying the observed effects, potentially through the incorporation of molecular studies investigating the interaction between the vectors and NSCLC cells.

We do not really follow this comment. The interactions between commonly used lentiviral vectors with VSV-G-carrying envelopes and human cells; and between adenovirus serotype 5-derived vectors and human cells are fully known. There is no new knowledge to gain about the molecular interaction of these viral vectors with surface receptors on the cancer cells and about their cell entry processes. In contrast, it was not known if these vectors could be delivered systemically via the CAM, how they subsequently distribute in the developing embryo and if they could reach human tumors growing on the CAM. We investigated this and provide methods to accomplish effective gene delivery to NSCLC-CAM tumors.

  1. The study does not provide a comparative analysis with other established preclinical models to demonstrate the added value or superiority of the NSCLC-CAM model in assessing new cancer therapies. A more extensive comparison with existing preclinical models for NSCLC is needed, along with a clear justification for the advantages of the proposed CAM model.

As explained in response to comment no.1, we do not propose to replace animal models by CAM tumor models. Hence, we do not wish to disqualify existing animal models by demonstrating superiority of the CAM tumor models. In our view, NSCLC-CAM tumor models provide a very useful intermediate step between in vitro cultures and animal models, reducing the number of animals needed in the final validation experiments. The clear advantages of the model are presented in our manuscript. They include reduced use of experimental animals; reduced costs; accelerated start of experiments (no ethical approval needed; no animal breeding delay; no cage adaptation period); simplicity allowing larger scale experiments (higher experimental group sizes or comparison of more experimental groups simultaneously); and shorter experiment duration. Together, this allows rapid selection of potential new cancer treatments for further evaluation of only the most effective ones in more demanding models.

  1. The study offers minimal discussion on the clinical relevance and applicability of its findings. A comprehensive understanding of how this preclinical model could be translated to human applications is pivotal for its acceptance in the scientific community. There is a notable absence of discussion on how the findings from this model can be translated to clinical settings, and how they align or contrast with existing knowledge and therapeutic strategies for NSCLC.

For our answer to this point, we refer to our answers above.

  1. A critical evaluation of the safety and potential toxicities of the systemic treatments and vector deliveries in the model is missing and needs to be addressed.

In our manuscript, we do not propose any systemic treatment for NSCLC. We established the NSCLC-CAM tumor models as tools for future therapy development experiments. As pointed out in our manuscript, we performed chemotherapy experiments to validate the NSCLC-CAM tumor models. To maximize relevance of this validation, we used a drug combination that is already in clinical use to treat NSCLC patients. Hence, there is no need to evaluate the toxicity of this treatment in a preclinical model, other than the safety dose-finding experiments we performed to make therapeutic experiments possible. The gene delivery vectors served to deliver the Firefly luciferase reporter gene. As expected, the vector delivery and transgene expression were non-toxic; there was no loss of embryos in vector-treated groups. Hence, our study suggests that viral vectors, in particular adenoviral vectors, can be used to test therapeutic gene delivery in the NSCLC-CAM tumor models. In such future studies, obviously careful evaluation of possible toxicities of transgene expression will be important. In this respect, it is useful that the transgene is not only delivered to cancer cells but also to non-malignant cells (see our answer to comment #3).

  1. The study lacks detailed information on adherence to ethical standards and guidelines regarding the use of animal models. It is crucial to ensure that all experimental procedures were conducted following relevant ethical norms. It would be pertinent to discuss any ethical considerations, particularly regarding the use of embryos, ensuring that animal welfare guidelines were followed.

As clearly pointed out in our manuscript, in the European Union (and many other countries) experiments in embryos are not considered animal experiments and do thus not require approval from an animal experimentation committee. Therefore, there are no standards or guidelines set for these experiments. Nevertheless, we of course adhered to our personal ethical norms. Therefore, hypothermia was applied before tissue dissection and euthanasia as described in our manuscript.

10.It is crucial to acknowledge the potential limitations and shortcomings of your approach, such as the differences between chicken embryonic and human environments and any potential biases in your study. A more detailed and thoughtful discussion on the limitations of the study, including potential biases and their impact on the results, is essential.

In regard to differences between chicken embryos and humans, we refer to our answer to comment 1. Limitations of the CAM tumor model are shared with other preclinical xenograft models. We feel that our discussion is balanced. We do emphasize the main limitation of the CAM tumor model, i.e., the short period available for measurement of therapeutic responses. This might underestimate measured responses and does not allow to ascertain durable responses.

Given the multiple areas of concern, substantial revisions and potentially additional experiments are required to address the shortcomings and validate the findings adequately. The paper must address these concerns with due diligence to elevate its scientific validity and translational potential.

As will be clear from the above, we disagree with most of this reviewer’s concerns. Many points were already addressed in our original manuscript. Other points we responded to above, clarifying them or explaining why we disagree. We have not revised our manuscript based on these issues.

Round 2

Reviewer 2 Report (New Reviewer)

I would to like to thank the authors for their pertinent response. Therefore, in my opinion, their work is suitable for publication in the present form.

Author Response

We thank the reviewer for the thorough review, which has helped us improve our manuscript.

Reviewer 3 Report (New Reviewer)

The 2nd revised article embarks on an interesting avenue for preclinical development for NSCLC. However, there are significantly notable gaps in the methodology's rationale, detailed quantitative data presentation, and translational potential.

1.     While the CAM model in chicken embryos is an accepted model for tumor growth, it differs from the human physiological environment. Factors like immune response, vascularization, and tissue interaction in the human context could lead to different results. It is essential to understand the limitations of the CAM model and not over-generalize findings.

2.     While five NSCLC cell lines were evaluated, only two (A549 and H460) were characterized and tested in depth. This narrow focus may limit the broader application of the findings.

3.     The histological differences observed between A549 and H460 CAM tumors need to be explored further. The implications of these differences in terms of therapy responsiveness, metastatic potential, and overall tumor biology are not discussed.

4.     A 100-fold increase in luminescence intensity with the adenovirus vector is mentioned, but the implications of this for tumor visualization, tracking, or therapeutic monitoring are not explored.

5.     The paper mentions minimal effects on embryo survival. However, it fails to provide specifics on the extent of mortality and morbidity, raising concerns about the ethical considerations of using this model if substantial embryo death or malformation occurred.

6.     The conclusion that the CAM tumor model provides a rapid evaluation for new cancer therapies is overly broad. To make such a claim, the study should have directly compared the CAM model's efficacy and speed with other widely accepted NSCLC models.

7.     The use of chicken embryos and the impact of xenografting on their survival and development raise ethical considerations. The study should address these concerns explicitly, clarifying any potential harm and justifying its methodology from an ethical standpoint.

This study, while presenting a novel approach to modeling NSCLC, lacks depth in exploration of the broader implications of its findings. To be of substantive clinical and scientific value, the research needs a more rigorous experimental design, a comprehensive exploration of results, and a more careful consideration of its ethical stance.

The 2nd revised article embarks on an interesting avenue for preclinical development for NSCLC. However, there are significantly notable gaps in the methodology's rationale, detailed quantitative data presentation, and translational potential.

1.     While the CAM model in chicken embryos is an accepted model for tumor growth, it differs from the human physiological environment. Factors like immune response, vascularization, and tissue interaction in the human context could lead to different results. It is essential to understand the limitations of the CAM model and not over-generalize findings.

2.     While five NSCLC cell lines were evaluated, only two (A549 and H460) were characterized and tested in depth. This narrow focus may limit the broader application of the findings.

3.     The histological differences observed between A549 and H460 CAM tumors need to be explored further. The implications of these differences in terms of therapy responsiveness, metastatic potential, and overall tumor biology are not discussed.

4.     A 100-fold increase in luminescence intensity with the adenovirus vector is mentioned, but the implications of this for tumor visualization, tracking, or therapeutic monitoring are not explored.

5.     The paper mentions minimal effects on embryo survival. However, it fails to provide specifics on the extent of mortality and morbidity, raising concerns about the ethical considerations of using this model if substantial embryo death or malformation occurred.

6.     The conclusion that the CAM tumor model provides a rapid evaluation for new cancer therapies is overly broad. To make such a claim, the study should have directly compared the CAM model's efficacy and speed with other widely accepted NSCLC models.

7.     The use of chicken embryos and the impact of xenografting on their survival and development raise ethical considerations. The study should address these concerns explicitly, clarifying any potential harm and justifying its methodology from an ethical standpoint.

This study, while presenting a novel approach to modeling NSCLC, lacks depth in exploration of the broader implications of its findings. To be of substantive clinical and scientific value, the research needs a more rigorous experimental design, a comprehensive exploration of results, and a more careful consideration of its ethical stance.

Author Response

We send our opinion on this review to the editor. We have not revised our manuscript.

This manuscript is a resubmission of an earlier submission. The following is a list of the peer review reports and author responses from that submission.

Round 1

Reviewer 1 Report

NSCLC is currently being treated with Immune check point Blockade(ICB), such as anti-PD1 and anti-CTLA4 in combination with other treatment including Chemotherapy. Chemotherapy has a role in immunomodulation of the Tumor microenvironment. Hence using CAM model will not be helpful in studying the mechanism of resistance to current therapy that could be in part due to Immune compartment modification toward resistance.  

Reviewer 2 Report

From a biostats and preclinical epidemiology point of view, I`ve some suggestions for the Authors:

- material and methods section must be reported before results, since it`s their background!

- material and methods section must be reported before discussion, of course!

- no statistical analysis plan, a full chapter has to be added, actually I can`t verify your approach

- no table 1, main cell lines/cultures characteristics; it`s absolutely mandatory

- no tables, results must be reported into tables and not into figures, as actually done

- I`m unable to understand on how many stats units your research has been conducted, this must be clearly stated in a fully detailed form

- power analysis, sigma parameter is lacking and it`s mandatory to perform such sample size estimations

- several time series have been estimated, there`s no description of their methodological approach

- moreover, mind that these topics must be statistically treated as “repeated, non independent measures”, there`s no comment about it

- the “supposed” KM curves in supplements can not be estimated with an extremely reduced number of events (which ones?), being totally unreliable